# Nonparametric Boundary Geometry in Physics Informed Deep Learning

**Scott Cameron**
Oxford University, Instadeep Ltd.
United Kingdom
scameron@instadeep.com

**Arnu Pretorius**
Instadeep Ltd.
South Africa

**Stephen Roberts**
Oxford University
United Kingdom

## Abstract

Engineering design problems frequently require solving systems of partial differential equations with boundary conditions specified on object geometries in the form of a triangular mesh. These boundary geometries are provided by a designer and are problem dependent. The efficiency of the design process greatly benefits from fast turnaround times when repeatedly solving PDEs on various geometries. However, most current work that uses machine learning to speed up the solution process relies heavily on a fixed parameterization of the geometry, which cannot be changed after training. This severely limits the possibility of reusing a trained model across a variety of design problems. In this work, we propose a novel neural operator architecture which accepts boundary geometry, in the form of triangular meshes, as input and produces an approximate solution to a given PDE as output. Once trained, the model can be used to rapidly estimate the PDE solution over a new geometry, without the need for retraining or representation of the geometry with a pre-specified parameterization.

## 1 Introduction

The prospect of applying machine learning to complex computational problems in science and engineering is appealing for a variety of reasons. In particular, surrogate models allow for rapid predictions to be made without performing lengthy and expensive computations. Surrogate models are especially beneficial in iterative design of technical systems that traditionally rely on sophisticated simulations. One challenge in such applications is the principled inclusion of inductive biases to preserve properties of the underlying physics. Various approaches have been proposed, such as directly parameterizing models to obey conservation laws [1; 10], or by the inclusion of constraints [22; 18]. Of key relevance to this paper is the approach popularly known as *Physics Informed Neural Networks* (PINNs) [12].

### 1.1 Physics informed neural networks

In their simplest form, PINNs are multilayer perceptrons trained to minimize a loss function based on the residual of some partial differential equation (PDE). The PDE in question is a system of equations, with solution $\phi$, defined on some open region $\Omega$ with boundary $\delta\Omega$:

$$F(\phi) := F(x, \phi, \nabla\phi, \ldots) = f(x) \quad x \in \Omega, \tag{1}$$

$$G(\phi) := G(x, \phi, \nabla\phi, \ldots) = g(x) \quad x \in \delta\Omega, \tag{2}$$

for some smooth $F, G, f$, and $g$. For the vast majority of realistic physical problems, the domain $\Omega$ is a subset of three dimensional Euclidean space $\mathbb{R}^3$. Here we shall furthermore assume that $\Omega$ is bounded and the boundary $\delta\Omega$ is compact. Many algorithms and techniques to solve such systems

37th Conference on Neural Information Processing Systems (NeurIPS 2023).

have been developed and extensively studied over the previous decades; perhaps the most popular approach being that of the finite element method (FEM), which approximates the solution piece-wise on a (volumetric) mesh discretization of $\Omega$. See Reddy [13] for an in depth introduction to FEM. The PINN approach approximates the solution $\phi$ using a smooth[1] neural network with parameters $\theta$. The PINN loss function measures the discrepancy of the network approximation from the PDE:[2]

$$\mathcal{L}(\theta) = \int_{\Omega} \left\| F(\phi_\theta) - f(x) \right\|^2 \lambda_f(x) \, \mathrm{d}x + \int_{\delta\Omega} \left\| G(\phi_\theta) - g(x) \right\|^2 \lambda_g(x) \, \mathrm{d}x \,. \tag{3}$$

Here, $\lambda_f$ and $\lambda_g$ are densities which may be chosen to increase or decrease importance of different points in the domain. In practice, these integrals are approximated using Monte Carlo sampling and the PDE residuals calculated using automatic differentiation.

Hypothetically speaking, with a universal function approximator, the global optimizer of Equation (3) should be able to provide an arbitrarily accurate solution to the the PDE. However, this is rarely realized in practice and one often needs to carefully tune the model and problem setup to give acceptable results. Various proposals have been made to improve the stability and robustness of PINNs such as enforcing exact boundary conditions using signed distance fields [16], highway networks [19; 15], alternative loss functions based on the variational formulation of PDEs [23], and causal training for time-dependent problems [20]. Furthermore, in industry applications, one would likely have access to (possibly sparse) simulation data, which would be used in a supervised fashion, with the PINN loss acting as a regularizer. For simplicity, in this paper, as a proof of concept, we will only consider a basic scheme for training PINNs without simulation data; however it should be noted that in practice more advanced methods would typically be required.

One considerable downside to PINNs is that the time taken to train the network may be comparable to or even significantly longer than applying classical techniques like FEM. Hence PINNs do not generally provide any advantage for one-off problems and traditional techniques provide better accuracy guarantees. However, it is frequently the case that one wants to solve a PDE for a family of conditions; for example by varying the parameters of the PDE or with a variety of different source terms. In these situations, PINNs can be trained conditionally to predict the solution of the PDE as a function of the source term or PDE parameters. These models learn a map from one function space to another and are thus called *neural operators* [6]. The advantage provided by neural operators lies in their capacity to make extremely fast predictions once they have been trained, whilst an FEM solver would generally have to start from scratch for each new variant of the problem.

One important aspect of modeling in science and engineering applications is the definition of boundary conditions. It is frequently the case in engineering design that one defines some geometry in computer-aided design (CAD) software, and subsequently runs an FEM simulation to calculate physical properties such as stress, heat transfer, or aerodynamics. After observing the results of the simulation, the designer may need to modify the geometry and repeat the simulation several more times. In such cases, the actual boundary conditions are often defined by the physics in question (perhaps based on material properties), but the geometry is modified between simulations. Hennigh et al. [3] provide an interesting example of a multi-physics design optimization problem involving a heatsink, where the geometry to be varied is described by a fixed set of parameters. They train a conditional PINN and optimize over the geometry using the trained PINN as a surrogate model for the heat transfer. This approach, however, requires a fixed parameterization of the geometry decided upon at training time and therefore the trained model is limited to this particular design problem.

In this paper, we propose a new neural operator architecture to learn the PDE solution operator, taking as input the boundary geometry in the form of a triangular mesh. This has many benefits:

- Once trained, the model may be used to rapidly predict the solution to the given PDE on new geometry without requiring expensive simulation.

- Since boundary geometry is provided as a triangular mesh, this allows the model to be reused across a wide variety of problems without retraining, and without requiring the geometry to conform to some fixed parameterization.

---

[1] i.e. of the same differentiability class as the PDE. For example, ReLU is not everywhere differentiable. Although it is weakly differentiable, its second derivative is almost everywhere zero, which makes it inadequate for general second order PDEs.

[2] Other norms may be used such as $L^p$ or Huber loss instead of $L^2$.

- A fast predictive model enables the use of numerical optimization in the design process, potentially allowing for improved productivity and reduced iteration time. In particular, since the neural operator is differentiable, gradient-based optimization can be used to optimize geometry.

- The increased applicability and utility of this type of model provides better justification for the time and resource costs involved in large-scale training in comparison to other PINNs which have limited reusability.

## 2 Geometry encoding with mesh-based neural operators

Solid geometry in CAD software is frequently represented by its boundary in the form of a triangular mesh. For physically realistic geometry, such a boundary mesh is always *manifold*. This means the mesh is closed (has no holes in the surface, nor a boundary), and each edge separates precisely two faces. Meshes which are non-triangular can always be triangulated, and we therefore restrict ourselves to this case. As a consequence of these properties, we may employ the architecture of MeshCNN [2]. MeshCNN may be thought of as a direct generalization of convolutional neural networks to non-Euclidian geometries. This approach exploits a key feature of triangular manifold meshes — that every edge has precisely four immediate neighbours: those of the two triangles connected by that edge.[3] Assuming we consider only positively oriented triangles (triangles whose normal vector points to the exterior of the enclosed volume), there are two possible orderings for the edges making up these two triangles. To enforce invariance to this ordering, Hanocka et al. [2] propose using feature vectors which are symmetric under a change of order. This is done by mapping the edge features of the four neighbours as follows

$$(e_1, e_2, e_3, e_4) \mapsto (e_1 + e_3, |e_1 - e_3|, e_2 + e_4, |e_2 - e_4|). \tag{4}$$

Let $\tilde{e}_{i,j}$ denote the above transformed feature vector for the $i^{\text{th}}$ edge for $j = 1, \ldots, 4$, and $\tilde{e}_{i,0}$ be the feature vector of edge $i$. The mesh convolution operation is then defined as

$$\text{conv}(K, \tilde{e})_i = \sum_{j=0}^{4} K_j \tilde{e}_{i,j}, \tag{5}$$

where each $K_j$ is a matrix of kernel weights. The fixed number of neighbours involved in the convolution greatly simplifies the implementation as compared to other neural network on graph-like domains. Hanocka et al. [2] also design a pooling operation on the mesh which collapses certain edges, in analogy with pooling operations typically found in convolutional neural networks; however, this mechanism is more complicated to implement and we instead use standard nonlinear activation functions as is also commonly seen in normal convolutional neural networks.

### 2.1 PDE solution operator

The solution to the PDE is a function of the boundary geometry. The dependency on the boundary geometry may be complex and highly nonlinear. The best method we have found to incorporate such dependence is in the use of the *attention* mechanism [17]. For this, we calculate encoded boundary geometry features using the MeshCNN. This takes into account local geometric information of the mesh. For the approximate PDE solution, we use a Transformer decoder architecture which uses cross-attention to aggregate these geometric features. Attention layers allow the model to take into account global geometric information, and allows the model to attend to different parts of the surface in different proportions depending on the point at which the function $\phi_\theta$ is being evaluated.

Importantly, as a deviation from the standard Transformer decoder, we do not include a self-attention layer over the points of the domain being evaluated. Naturally, the value of $\phi_\theta$ at a point $x$ should be independent of whether or not one is simultaneously calculating its value at another point $x'$. In other words, the forward pass of the neural network should give identical results, regardless of whether it is performed over the domain one point at a time, or performed over those points in parallel as a batch. This can be stated mathematically with the chain rule: $\frac{\partial \phi(x)}{\partial x'} = \frac{\partial \phi(x)}{\partial x} \frac{\partial x}{\partial x'} = 0$. This would no longer be the case if self-attention layers were included in the decoder.

---

[3]This is analogous to the von Neumann neighbourhood in cellular automata as opposed to the Moore neighbourhood.

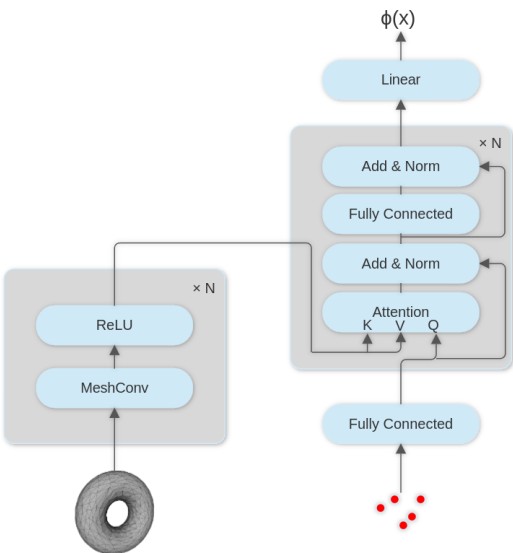

Figure 1: Neural operator architecture.

Barring the lack of self-attention, our decoder is largely the same as that given by Vaswani et al. [17]: a sequence of blocks, each of which consists of multi-head dot-product attention, with a residual connection and layer normalization, followed by a single hidden-layer feed-forward neural network, also with a residual connection and layer normalization. In the attention layers, the query vectors are linear projections of the outputs from the previous layer, while the key and value vectors are linear projections of the encoded edge features which are the output of the MeshCNN. The final layer output is just a linear layer which projects down to the dimension of $\phi$. Figure 1 gives a visual representation of this architecture.

We attempted many variations in building up to this architecture. One specific approach we tried was to use an operator of a similar form to DeepONet [9]. DeepONet is a neural operator defined by two neural networks; the *branch* which takes as input point evaluations of a function at some predefined locations, and as output produces the final layer weights for the second network, the *trunk*, which itself is the result of applying the operator. Lu et al. [9] show that under mild assumptions (such as non-polynomial activation functions), any continuous nonlinear operator can be approximated arbitrarily accurately on a compact set by a neural operator of this form. Our first approach was to use DeepONet, using a MeshCNN as the branch network and simply taking the mean over edges to reduce its output to a fixed dimensionality. Despite its theoretical properties, it soon became clear that, in practice, this model did not provide enough flexibility for this application.

As an improvement, we thereafter tried a single attention layer to perform a weighted average over MeshCNN output, which seemed to help somewhat, but still failed to give any results of satisfactory accuracy. This led us to consider using multiple layers of attention, which showed a significant performance increase and proved to be flexible enough to model the PDE solution operator accurately.[4]

## 2.2 Invariance to geometry preserving transformations

In a similar manner to standard convolutional neural networks' invariance to translations, MeshCNN is invariant to transformations which preserve the local topology of the mesh. When the input features of the mesh depend on local geometry (such as curvature, torsion, or a given vector field) then MeshCNN is invariant only to transformations which preserve this local geometry. This invariance property is manifest in the definition of the convolution in Equation (5). Ideally, for application to boundary geometry, our model would also be invariant to transformations of the continuum manifold which preserve local geometry; however, the discretization of the continuum manifold into a trian-

---

[4]While the architecture is flexible enough to give accurate PDE solutions, this does not guarantee that simply training on Equation (3) will achieve an accurate fit in practice without more advanced schemes.

gular mesh breaks this invariance, and so the MeshCNN is not guaranteed to be truly invariant under different mesh discretizations. Despite this, Hanocka et al. [2] report that, in experiments, this invariance is approximately preserved, and trained MeshCNNs may give quantitatively similar results on two different triangulations of the same mesh. Importantly, this discretization should be done in a way which best preserves the overall geometry and maintains a fine enough resolution in order to be valid. Geometric features are calculated in a manner which approximates their continuous counterparts in the limit of an infinite resolution mesh. Naturally, a poor quality discretization (with either a very low resolution or one which poorly approximates the geometry) should be expected to adversely affect the quality of the results. Approximate invariance to discretization depends on the assumption that the geometric features themselves are approximately invariant to the discretization process, which is only true for meshes of sufficiently fine resolution. The extent to which the boundary mesh resolution can be reduced without significant loss of quality in model predictions poses an interesting research question, albeit one which we leave for future work.

To maintain the appropriate invariance properties of the PDE solution map, edge features on the mesh must be chosen accordingly. For mesh segmentation and classification purposes, Hanocka et al. [2] employ features which are invariant to similarity transformations, such as translation, rotation and scaling. This is appropriate because the predictions of the network transform as scalars. In our case, the mesh is assumed to be embedded in $\mathbb{R}^3$ and the output of the network is a function on the interior domain which is a subset of the same $\mathbb{R}^3$. Thus applying a transformation $T$, which is a symmetry of the PDE, to the boundary geometry, should result in the same solution with the inverse transformation applied to the interior domain $\phi(x) \to \phi\big(T^{-1}(x)\big)$.[5] The solution operator is thus *contravariant* as opposed to invariant under symmetry transformations. Although it is not obvious how this contravariance could be explicitly encoded in the operator network architecture, we can encourage the network to learn contravariance by using covariant or contravariant edge features, and data augmentation by random transformations of the boundary geometry during training. As examples of such edge features, we use the midpoint and length of each edge, local surface normal vectors, and local curvature.

## 3 Experiments

For boundary geometry training data, we use a collection of meshes created in Fusion 360 [21].[6] After preprocessing and filtering, our data set contains about 12 thousand triangular meshes. For validation data, we use an FEM solver on a small collection of meshes, which we treat as our ground truth. While FEM solvers are only approximate methods, in the cases we consider they are extremely accurate and so their errors are negligible. We measure the mean squared error (MSE) of our neural operator approximation compared to the FEM solution and report these errors as percentages of the mean square value (the average intensity) of the FEM solution. For a comparison of our model to vanilla PINNs trained on individual meshes, see Appendix A. During training, we save the model parameters with the lowest average MSE over the validation set.

We generate geometry-encoding edge features as described in Section 2.2, consisting of the edge midpoint, length, normal vector, and scalar curvature. Interior and boundary points are generated by sampling uniformly inside and over the surface respectively. These points are used to estimate the loss function in Equation (3), where we replace the $L^2$ norm with a Huber loss for stability. We use a data augmentation scheme, consisting of random translations, rotations, and nonuniform scaling of the meshes.

For our network architecture, we use a MeshCNN with ReLU[7] nonlinearities, and SiLU nonlinearities for all activation functions in the Transformer decoder. As input to the Transformer decoder we found that a (learnable) linear combination of Fourier features and SiLU embeddings worked better than either alone. We also attempted to use only Fourier features as input to the decoder but found that it gave considerably worse results. Our MeshCNN uses four convolutional layers, each having 512 output channels. For the Transformer decoder, we use 5 blocks, using attention layers

---

[5]Here we are assuming the value of $\phi(x)$ transforms as a scalar on the domain. Similar notions of covariance/contravariance apply when $\phi(x)$ transforms as a tensor.

[6]Available at https://github.com/AutodeskAILab/Fusion360GalleryDataset.

[7]The smoothness requirement for PINNs only applies to the part of the network being differentiated, i.e. the decoder which takes $x$ as input should be smooth but the MeshCNN, which does not depend on $x$, can have arbitrary nonlinearities.

with 8 heads and a feature size of 512. The feed-forward network sublayers in each block have a hidden-layer size of 2048.

We use stochastic gradient descent with momentum as we found other optimizers (Adam [5], RAdam [8] and RMSProp) to be very unstable. We use a cosine annealing learning rate schedule with a linear warmup to a maximum learning rate of $0.001$. To improve stability we use gradient clipping and gradient accumulation to increase the effective batch size.

For the experiments below, unless otherwise stated, we use a Gaussian source term centered at the origin as a smoothed-out approximation to a point-like source. During data augmentation, we ensure that the origin point is always on the interior of the PDE domain ($0 \in \Omega$).

All experiments use the same network architecture and hyper-parameters. They were trained on an RTX 3070 with 8Gb of VRAM.

## 3.1 Poisson equation

The Poisson equation with Dirichlet boundary conditions is defined as

$$-\nabla^2 \phi = f(x) \quad x \in \Omega, \tag{6}$$
$$\phi(x) = 0 \quad x \in \delta\Omega. \tag{7}$$

This equation arises in a number of applications including steady state diffusion processes, Newtonian gravity and the study of electromagnetism.

For this equation, as a sanity check, we first use the so-called "method of manufactured solutions" to compare against a simple analytic solution. By substituting in the analytic solution: $\phi = \cos(x)\sin(y)$, we arrive at the Poisson equation with the appropriate source term and boundary conditions as follows:

$$-\nabla^2 \phi = 2\cos(x)\sin(y) \quad x \in \Omega, \tag{8}$$
$$\phi(x) = \cos(x)\sin(y) \quad x \in \delta\Omega. \tag{9}$$

This method is commonly used to test numerical solvers against known analytic solutions. Figure 2 shows a comparison between the manufactured solution and the neural operator prediction on an example test geometry. In all figures below, the color intensities of the FEM or true solution and those of the neural operator share the same normalization. In this example, the MSE is 4% of the average intensity, and the model prediction had an average error of 5% across the validation set.

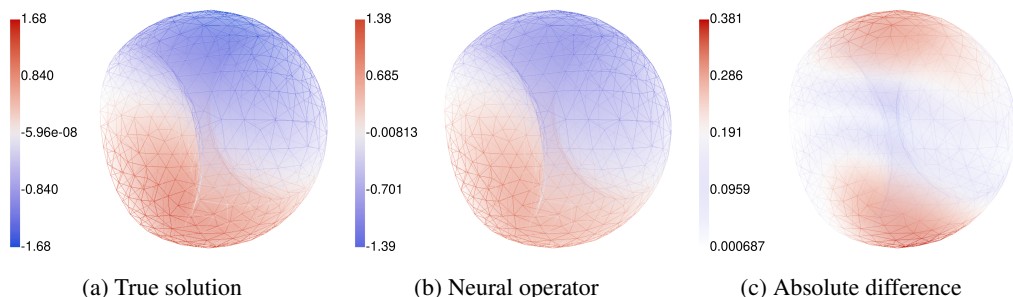

| (a) True solution | (b) Neural operator | (c) Absolute difference |

Figure 2: Poisson equation: manufactured solution. a shows the manufactured analytical solution, b shows the output of our trained model on the same geometry, and c show the absolute error of the model compared to the true solution.

For the Gaussian point-like source term, Figure 3 shows the comparison of the neural operator prediction to the FEM solution on two test meshes, having an MSE of 3% and 8% of the average intensity respectively.

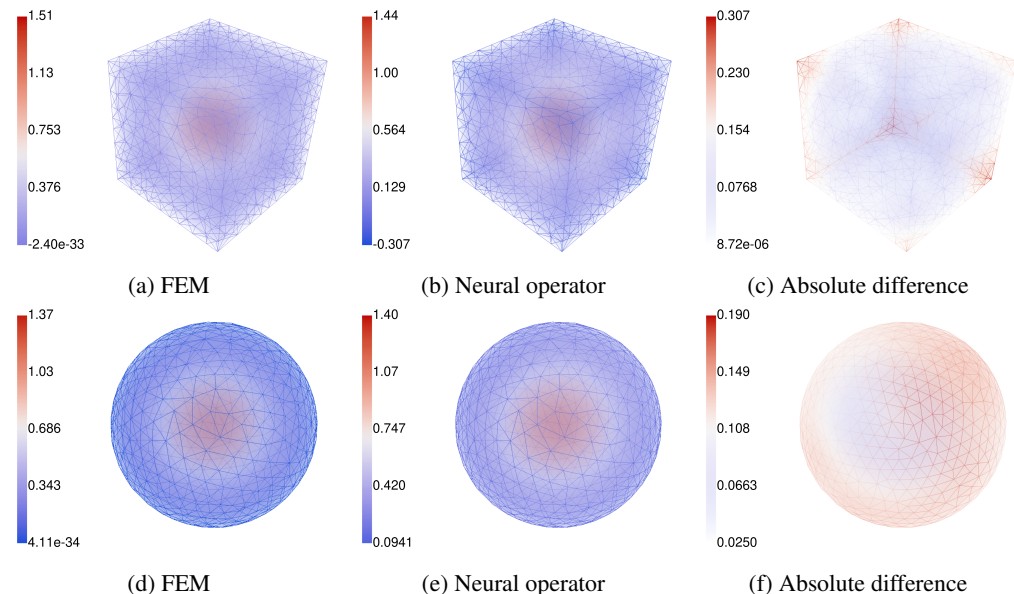

|   |   |   |
|---|---|---|
| (a) FEM | (b) Neural operator | (c) Absolute difference |
| (d) FEM | (e) Neural operator | (f) Absolute difference |

Figure 3: Poisson equation. a, d show the FEM solution, b, e show the output of our trained model on the same geometry, and c, f show the absolute error of the model compared to the FEM solution.

## 3.2 Helmholtz equation

The Helmholtz equation is the eigenvalue equation for the Laplace operator. The Helmholtz problem with Neumann boundary conditions is given as:

$$\nabla^2 \phi + \kappa^2 \phi = f(x) \quad x \in \Omega, \tag{10}$$
$$n \cdot \nabla \phi(x) = 0 \quad x \in \delta\Omega, \tag{11}$$

where $\kappa^2$ is the eigenvalue and $n$ is the unit normal vector to the surface $\delta\Omega$ at $x$. The solutions to this equation form a basis which diagonalizes the Laplace operator, and hence this equation arises in many applications. In particular, this equation is used in studying wave propagation, and therefore specifically in acoustics and optics. Here we assume the eigenvalue of interest to be $\kappa^2 = 1$.

Figure 4 shows the FEM solution and neural operator prediction for the Helmholtz equation. On both these test meshes the MSE of the neural operator is approximately 2% of the average intensity.

## 3.3 Steady state reaction-diffusion system

Reaction-diffusion equations are frequently used in modeling chemical reactions, biological systems such as in predator-prey models, combustion, population dynamics, and spread of disease in epidemiology. We consider the case of a quadratic reaction term, known as Fisher's equation, with Neumann boundary conditions in the stationary ($\partial_t \phi = 0$) limit:

$$\nabla^2 \phi + \phi(1 - \phi) = f(x) \quad x \in \Omega, \tag{12}$$
$$n \cdot \nabla \phi(x) = 0 \quad x \in \delta\Omega. \tag{13}$$

Figure 5 shows our neural operator predictions on three test meshes. The MSEs for these predictions are 1%, 1%, and 5% of the mean intensities respectively. There is a noticeable qualitative difference between the neural operator solution and the FEM solution in the third row (Figures 5g to 5i). The small relative volume covered by this error suppresses its contribution to the MSE, and to the loss function in Equation (3).

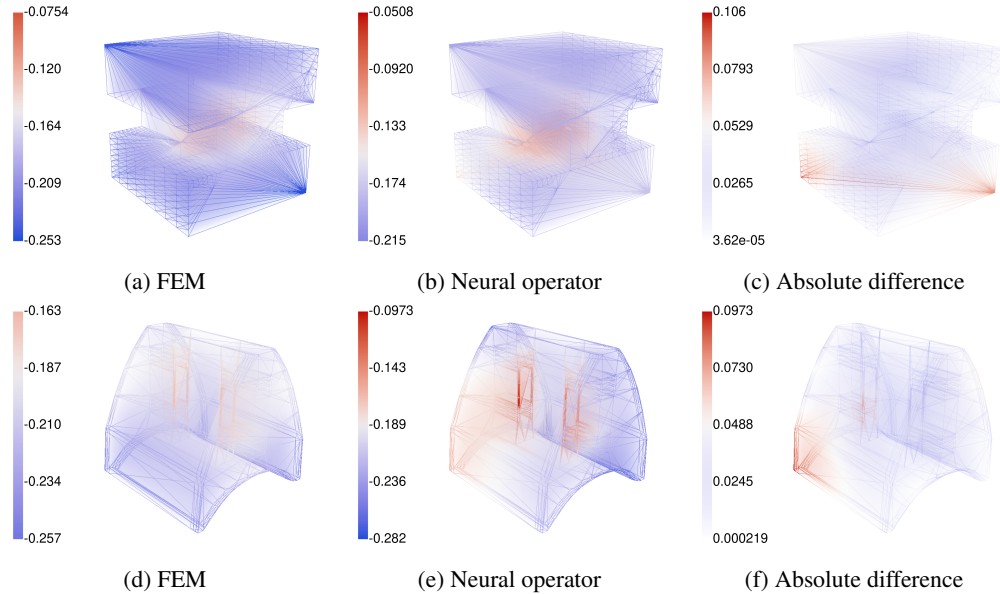

(a) FEM      (b) Neural operator      (c) Absolute difference

(d) FEM      (e) Neural operator      (f) Absolute difference

Figure 4: Helmholtz equation. a, d show the FEM solution, b, e show the output of our trained model on the same geometry, and c, f show the absolute error of the model compared to the FEM solution.

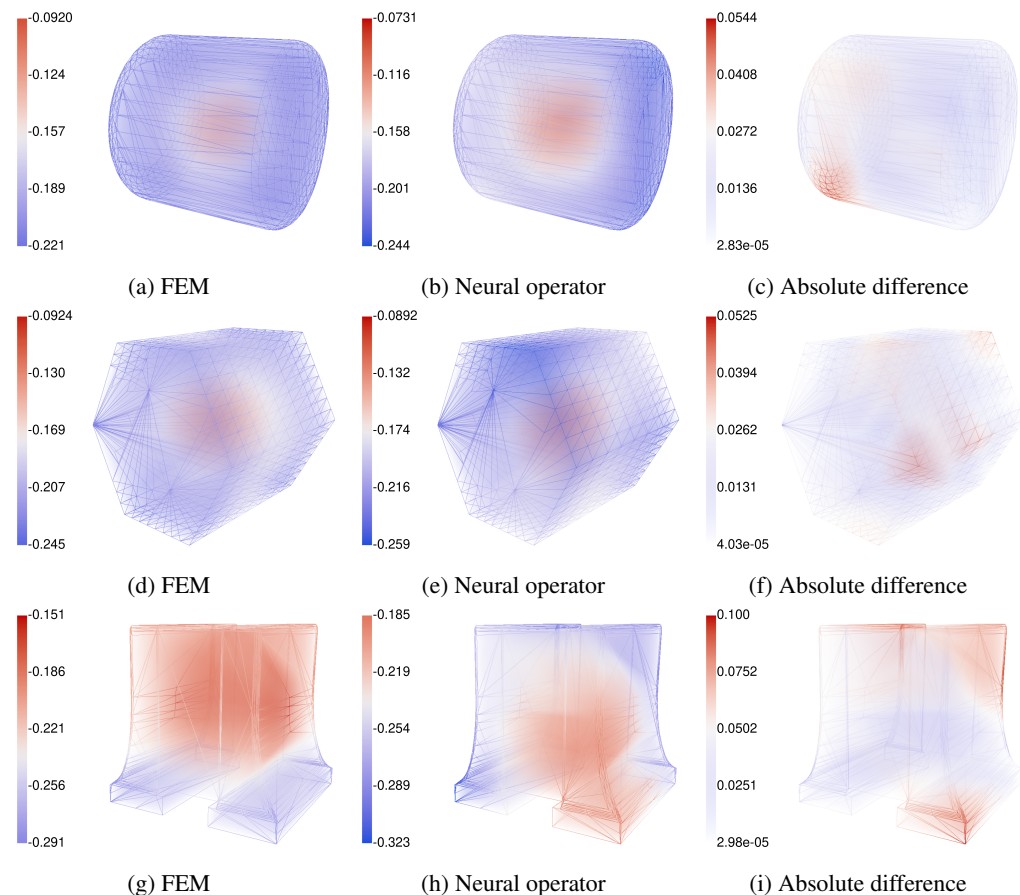

(a) FEM      (b) Neural operator      (c) Absolute difference

(d) FEM      (e) Neural operator      (f) Absolute difference

(g) FEM      (h) Neural operator      (i) Absolute difference

Figure 5: Reaction–diffusion system. a, d, g show the FEM solution, b, e, h show the output of our trained model on the same geometry, and c, f, i show the absolute error of the model compared to the FEM solution.

# 4 Related work

There have been many advances in physics informed learning in recent years. However, relatively few of these advances specifically focus on boundary geometry.

On fixed rectangular geometries, convolutional neural networks and Fourier neural operators [7] can be used to predict solutions to PDEs as a function of the source term, or to solve inverse problems such as inference of initial conditions of dynamical field from its current configuration. They may be trained with or without a physics informed approach. However, the these models cannot be used on irregular domains.

Using graph neural networks, one can extend these approaches to irregular domains using a volumetric mesh. Pfaff et al. [11] build a model to predict the result of an expensive simulation on a mesh, by training on data generated by the simulation. However, this approach does not encode physical inductive biases as PINNs do, using a standard supervised approach. Hamiltonian graph neural networks [14] can be used on meshes similarly, with the difference that they explicitly encode energy conservation by simulating a Hamiltonian system.

Kashefi and Mukerji [4] have proposed a model to approximate the solution to a given PDE on multiple irregular geometries. Their model is trained in a physics informed manner and is based on PointNet, which is a neural operator designed for processing point clouds. These point clouds form an implicit representation of the geometry and their model can be used to make predictions on new geometries just by generating interior samples. However, compared to point clouds, meshes contain far more geometric information [2] such as local connectivity and curvature.

# 5 Conclusion

In this paper, we propose a novel neural operator architecture for the purposes of learning PDE solutions as a function of boundary geometry without restricting the geometry to a fixed, finite-dimensional parameterization. This provides a variety of benefits and significantly extends the possible applications and reusability of trained models over other PINNs. We believe this is an important step toward feasible PINN technology at industry scale, and reducing time and resource costs associated with traditional simulations.

We showed that our neural operator is capable of approximating the PDE solution operator, even when trained without simulation data or more advanced PINN methodologies. While we chose not to incorporate more sophisticated training techniques for the sake of simplicity and proof of concept, we highly recommend that they be used in any practical application, along with example data from simulations to improve the reliability and robustness of the model. We believe this is especially important as the geometry and domain of interest increase in complexity.

## 5.1 Limitations

There are a variety of limitations with using neural networks to approximate solutions to PDEs. The subject itself is still relatively immature when compared to traditional methods like FEM or finite volume methods, and lacks the theoretical guarantees and wealth of insight that have been developed for these methods. Problems of exploding or vanishing gradients are exacerbated by the triple, or even higher order derivatives appearing from the optimization of Equation (3). When training on the PDE itself, loss curves may vary by multiple orders of magnitude from one epoch to the next, sometimes increasing even when the error of the solution is decreasing. This makes the learning curves very difficult to interpret. Furthermore, small loss values do not necessarily mean that the solution itself is accurate, and this complicates error estimation and uncertainty quantification.

For our particular experiments, we found that the lowest errors for each validation mesh occurred at different epochs and so while we were able to get very accurate results (occasionally even below 1% MSE) for each individual validation mesh, if we monitored its error and used early stopping, we were not able to achieve such accurate results on each mesh simultaneously. We believe this is mostly due to the difficulties of fitting to the PDE directly rather than a limitation of the model itself, and we expect to achieve better results with supervised learning on data from simulations in addition to the PINN loss function.

## Acknowledgments and Disclosure of Funding

This work was sponsored by InstaDeep Ltd.

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

# A  Vanilla PINN results

For a few selected test meshes, we attempted training a standard PINN, without conditioning on the geometry. Here we show a comparison of these PINNs to our trained neural operator. The training code is identical for both the operator and standard PINNs, with the only differences being that the PINN model does not include any mesh encoding or attention layers, and the dataset is restricted to only a single mesh.

The PINN model architecture consists of 5 blocks, each of which is a fully connected feed-forward network with a skip connection and SiLU nonlinearities. The inputs and outputs of each block are of size 512, and each feed-forward network has a hidden-layer size of 2048. We use the linear combination of Fourier features and SiLU embeddings in the first layer. These parameters are chosen to match our neural operator for fair comparison. This model has 12.6 million parameters.

In all cases, the standard PINN fails miserably, producing far less accurate results than the neural operator. It appears that in this case, the process of learning on all geometries simultaneously improves the ability of the model to find an accurate solution to the PDE. Further investigation would need to be done to assess the extent to which this behaviour occurs, but we leave that for future work.

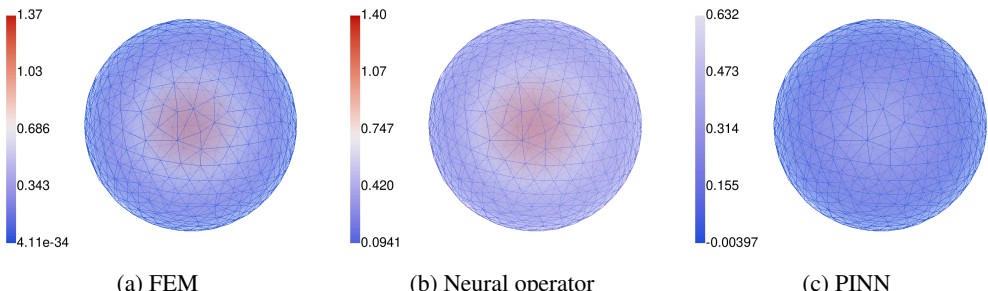

(a) FEM                    (b) Neural operator                    (c) PINN

Figure 6: Comparison to vanilla PINNs on the Poisson equation

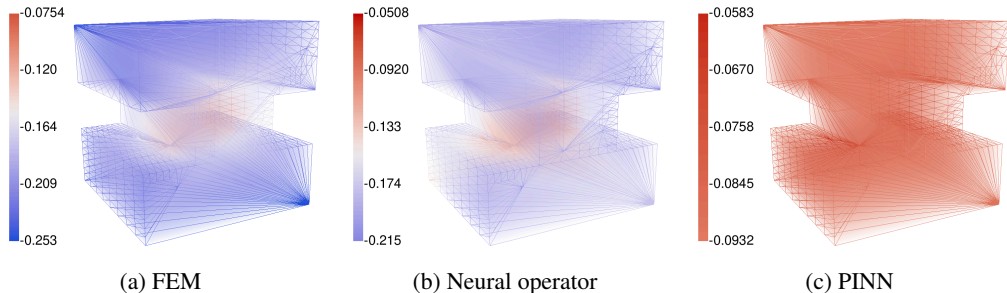

(a) FEM                    (b) Neural operator                    (c) PINN

Figure 7: Comparison to vanilla PINNs on Helmholtz equation

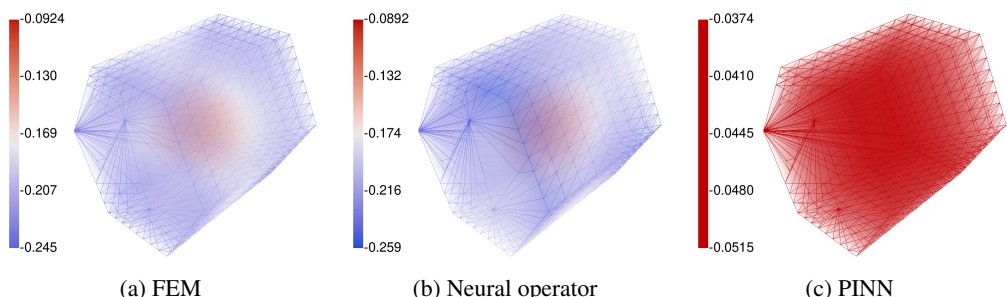

(a) FEM                    (b) Neural operator                    (c) PINN

Figure 8: Comparison to vanilla PINNs on reaction diffusion equation

