# OpenReview forum: "Nonparametric Boundary Geometry in Physics Informed Deep Learning"
_NeurIPS.cc/2023/Conference — NeurIPS 2023 poster_

### Official Review · Reviewer_mGk1 · 2023-07-04

**Soundness:** 3 good
**Presentation:** 3 good
**Contribution:** 3 good
**Rating:** 7
**Confidence:** 4

**Summary:**

The Authors introduce a cross-attention (without self-attention)  based decoder architecture for neural operators for solving PDEs. The architecture contains an encoder component that has the boundary defined by a triangular mesh as an input. The mesh encoder uses graph convolutions on the edges with uniform 4-nearest neighbour architecture across  the triangular mesh to define the features of the boundaries, the decoder attention layers propagate the information to create the PDE solution in the interior domain. The decoder uses ReLU, the transformer SiLU non-linearities. The latter to make sure the automatic differentiation for the PDE solution is smooth enough everywhere.

The system is trained with 12000 example meshes (geometries), validated by FEM solutions. The lowest validation error parameters from each epoch is used.  The methods is demonstrated for a set of simple geometries and equations with mostly encouraging results.

The capability of solving PDEs with varying boundaries in an efficient was is valuable for many industrial applications, and is an impactful addition to the  physical system simulation toolbox.

**Strengths:**

The introduction is clear and easy to follow, also  the manuscript brings clearly up the benefits of using PINNS in the industrial domain.

The neural network operator that can take a triangularisation define boundary as an input for the trained system, and hence allows for solution of gPDEs with geometries using only an inference of a neural network, with convolution and a layers of cross-attention has the potential to provide solution much faster than traditional  FEM solvers.

**Weaknesses:**

The architecture of the solution, especially its hyper parameters are not described.  Figures showing the components of the architecture are missing, and verbal explanation with references to similar architectures MeshNet and decoder from the original attention is all you need makes it hard to read. As introducing the novel architecture is the main result of the paper, one should concentrate on describing it in simple visual manner that allows a wider audience to grasp the essential.

Also, regularisation of PDE solutions when geometries are given with triangularisation have singularities. The manuscript should address this.

**Questions:**

Poisson  equation solutions have corner singularities in certain geometries. Now, analytically the PDE solution of triangulation based geometric behave badly on all vertices of the boundary mesh. It is possible to solve the problem by smoothing the surface, but this would require a different parametrisation of the surface, for example splines - as often used in industry. How would the current neural operator architecture address this problem?

---

> ### Author Rebuttal · Authors · 2023-08-10
>
> Please see our other response and the attached diagram for an explanation of our architecture. We hope this provides a clearer picture of our model.
>
> It is true that non-smooth boundaries (eg. sharp points or corners) can lead to PDE solutions which are non-differentiable, discontinuous or even have singularities. Non-smooth behavior is an important consideration for PINNs and they usually perform very poorly in these cases.
>
> There are at least 2 separate scenarios to consider here. (1) The geometry of interest is smooth, and the triangular mesh is a non-smooth approximation to the geometry, and (2) the geometry of interest does in fact have important non-smooth points such as corners.
>
> In the first case, we assume the true solution to the PDE is smooth, and so it may be that the PINN doesn’t have any trouble, since it is smooth by design. Essentially, the PINNs inherent smoothness naturally regularizes it. Even so, it may be valuable to reparameterize the boundary geometry with spline patches instead of flat triangles as is done in many FEM applications. The structure of this is still a mesh, but with additional information about the spline curvature. There is nothing to prevent this information from being used as feature information in the input to the MeshCNN. The MeshCNN doesn’t care whether the edges it is given are flat or curved, only the connectivity matters. Of course, during training, boundary and interior points still need to be sampled accordingly.
>
> In the second case PINNs are generally incapable of exhibiting this kind of non-smooth behavior without specific modifications. This is an important and rich line of research, and there are many ideas that address it; however, it is somewhat orthogonal to the points in our paper. We do mention some of this work in the introduction, but we do not ourselves directly address this point.

---

> > ### Comment · Reviewer_mGk1 · 2023-08-15
> >
> > Thank you for the authors for the clearer description of the structure of their solution. I wait to see this in the new version of the manuscript.
> >
> > You may also consider comparing your solution to BEM, as it has similarities. With the knowledge of the solution on the boundary one can use the Kirchhoff integral to calculate the inside provided that the medium is homogenous. Calculating this single convolution on the GPU may provide a faster inference, after the boundary values have been computed. In industrial applications the same equation may be solved repeatedly for different interior points.
> >
> > The boundary value consistency requirement in BEM is essentially a full "attention" to all other boundary values. With fast multipole expansions the contributions can be lumped up to significantly speed up the solution, in practise corresponding a convolution with a small kernel.

---

> > > ### Author Response · Authors · 2023-08-21
> > >
> > > Thank you for this suggestion.
> > > This is a very interesting point. The BEM certainly shares parallels with our operator approach and it may be a valuable discussion point to add to the paper given the space. This closely relates to the topic of learning Green's functions with neural operators.

---

### Official Review · Reviewer_4pp2 · 2023-07-04

**Soundness:** 3 good
**Presentation:** 2 fair
**Contribution:** 3 good
**Rating:** 5
**Confidence:** 3

**Summary:**

This paper proposes a neural operator methods that can take different boundary geometry, in the representation of a mesh, as input to solve different PDEs. To the best of my understanding, the proposed method can take geometry represented in different triangular meshes as input and predict PDE solutions (i.e. a function of the geometry). The proposed method used MeshCNN to extract geometry features, and use cross attention transformer as the decoder to produce the solution. The proposed model needs to be trained only once for a specific PDE, then it can be used to produce solution for different conditions.

This work is trying to address a very important problem, but due to the presentation issue and some concern with the results, I am leaning toward rejection. I think the paper can be greatly strengthen with an iteration with better presentation and more evaluation.

**Strengths:**

1. I think the problem this paper set-up to solve is of very important. When the engineers are using a model to speed up their simulation pipelines, they usually require solving the same problem with different geometries. If the trained model can only work on one specific geometry, then it might not be the best fit of such pipeline. With this said, if this paper is successful, it can enable neural operator to be used in more practical scenario.
2. The proposed method (if I understand it correctly) can indeed end handle different geometry while maintaining respected invariances required for the problem set-up.

**Weaknesses:**

1. Presentation issues. I found the paper (especially the method section) difficult to understand as it lacks of rigorous definition of the problem set-up (e.g. input/output of the model; how do we represent it; what’s the network architecture;). As the paper is proposing a specific way to achieve this goal, I think including a more rigorous definition of the model can improve reproducibility and help understand why specific model design choice is necessary.
2. The results does not necessarily support the claims. A important claim of the paper is that their proposed model can be trained over different geometries of the same PDE, thus provide advantage by amortizing the training compute to benefit multiple simulation rounds. To show this, I think it’s necessary to compare to the baseline where for each geometry, we train a neural operator (which we might not be able to do without data) or we solve for a PINN for each geometry (this is valid as PINNs can be solved for scratch). I expect this comparison turns out to be that the proposed method is comparable with the baseline, while being faster since PINNs requires optimization while the proposed method requires only forward pass. Right now the results only show comparison with the ground truth, which makes it difficult to see 1) whether the proposed method is accurate enough, and 2) whether the hypothesized advantage in amortizing the training across different geometry exists.
3. Agnostic to discrete action. While the paper cite MeshCNN to argue that in practice the feature extraction pipeline is not adversely affected by the discretization quality, I think this argument is not sufficient as simulation using FEM usually requires a careful discretization and it’s simulation is different from the task MeshCNN were designed to do (classification and segmentation). The fact that the method section mentioned the manifold assumption in L88-100 strengthen my concern. While this can be better fit into the future-works scope, I would recommend the authors to stress-test how different discretization affects the performance, and probably discuss how this method can be applied to non-curated meshes such as mesh obtained from real-world scans.

**Questions:**

See the weakness section.

**Limitations:**

Yes, the author has a good discussion of the limitation.

---

> ### Author Rebuttal · Authors · 2023-08-10
>
> Please see our other response and the attached diagram for an explanation of our architecture. We hope this provides a clearer picture of our model.
>
> We have added in the attached pdf two examples of vanilla PINNs trained on test geometries for comparison with our neural operator. For the architecture, we use the same feed forward layers that appear in our operator with silu non-linearities and skip connections. In other words, we train a PINN consisting of 5 blocks, where each block is of the form:
> $h_{t+1} = \operatorname{LayerNorm}(h_t + W_t \operatorname{silu}(U_t h_t + a_t) + b_t)$,
> where each $h_t$ is 512 dimensional, and each block has 2048 hidden neurons.
> These values are chosen in an attempt to make a fair comparison.
> Even though we observe the losses converging, the fitted model is often quite different from the ground truth solution. Without more sophisticated training techniques or simulation data (see the introduction of our paper) we have thus far been unable to attain better performance with vanilla PINNs on these geometries than what is shown in the attached pdf.
>
> The point regarding the discretization quality of the boundary is a very important one. The MeshCNN paper claims that in practice they noticed their model was able to maintain accurate predictions on different discretizations of the same object. However, it is important to note that these different discretizations were done in a way which preserves the overall geometry (this  requirement is much more restrictive on surface meshes than on volumetric meshes), while also maintaining similar resolution. Geometric features are calculated in a manner which approximates their continuous counterparts in the limit of an infinite resolution mesh. It does make sense then that a poor quality discretization, with either a very different resolution or one which poorly approximates the geometry should indeed adversely affect the quality of the results. Approximate invariance to discretization depends on the assumption that the geometric features themselves are approximately invariant to the discretization process, which is only true for meshes of fine enough resolution.
> As you mention, we will leave a rigorous assessment of mesh discretization to future work, but we will ensure that this point is more clearly discussed in the paper.

---

> > ### Comment · Reviewer_4pp2 · 2023-08-21
> >
> > Thanks the authors for the detailed response. I think a more detailed discussion of the point about discretization will help improve the manuscript.
> >
> > The results comparing with PINNs can greatly strengthen the paper - I encourage the authors to make such comparison more rigurous in the next revision.
> >
> > I'm happy to raise the score under the faith that the authors can include a better PINNs comparison, revise the writing to make the paper more clear, and include the discussion on surface discretization.

---

> > > ### Author Response · Authors · 2023-08-21
> > >
> > > Thank you for your feedback. We will certainly expand on the points regarding surface discretization and add the comparison to vanilla PINNs in the revised version of the paper.

---

### Official Review · Reviewer_ECF7 · 2023-07-06

**Soundness:** 2 fair
**Presentation:** 2 fair
**Contribution:** 2 fair
**Rating:** 5
**Confidence:** 2

**Summary:**

Physics Informed Neural Nets (PINN) have gained considerable attention lately. However, they are significantly expensive as compared to FEM or other classical PDE solvers. Moreover, any trained PINN is specific to the object geometry it has been trained upon. This paper proposes a solution to reuse the trained model to various object geometries by learning a PDE solution without restricting to a certain object parametrisation. It takes into account the bounding conditions on the object in terms of meshes and deploys MeshCNN instead of regular CNN to encode local geometric properties under an attention mechanish [17] to learn the solutions to PDE.

**Strengths:**

The proposed method addresses an important problem of making the physics-based learning more generic.

The paper discusses an important aspect of learning on meshes: contravariance/invariance of geometric properties under mesh formulations. It enforces the preservation of geometric quantities in order to solve the PDE.

The experiments show various well-known PDE systems are solved quite accurately using the proposed approach.

**Weaknesses:**

The writing is a bit difficult to follow. The use of footnotes is weird and interrupts the reader's flow.

Some design aspects of the paper are not clear. Lines 118-120 suggest that the self-attention block of the Transformer is removed.  The authors argue that the value of learned parameter at a point x should be independent of whether or not one is simultaneously calculating its value at another point x′. This argument needs to be explained further as geometrically close points manifest similar properties and learning them together (or simultaneously) can be beneficial to obtain the solution to the PDE. It is understandable that the geometrically distant points need not to considered simultaneously. Choosing to remove the self-attention layer seems to be a compromise and should be explained in more details.


**Questions:**

see weakness section

**Limitations:**

The limitations have been discussed

---

> ### Author Rebuttal · Authors · 2023-08-10
>
> Please see our other response and the attached diagram for an explanation of our architecture. We hope this provides a clearer picture of our model.
>
> We have attempted to explain above why it is undesirable for information to be shared between different points. For a bit of further clarification, it is certainly possible to build a model which does explicitly share information using eg. self-attention, but this would be a different species of operator. If $\mathcal{M}$ is the space of 2 dimensional closed boundary manifolds in $\mathbb{R}^{3}$ and $C^{2}(\mathbb{R}^{3})$ is the space of twice differentiable functions, our operator has type $\mathcal{M} \to C^{2}(\mathbb{R}^{3})$. By partial application of the MeshCNN part of the model, one arrives at a function $\phi \in C^{2}(\mathbb{R}^{3})$, which can be evaluated independently at any point within the domain. On the other hand, if we were to allow mixing of information between points with self-attention, we would have an operator of type $\mathcal{M} \to (\mathbb{R}^{3n} \to \mathbb{R}^{n})$ for all values of $n$. Partially applying the MeshCNN in this case would result in an object which is not an element of $C^{2}(\mathbb{R}^{3})$, but is itself an operator on a more complex space: something like $\hat\phi(x_1,x_2,\ldots) = (\phi(x_1), \phi(x_2), \ldots)$. In this case, the Jacobian of $\hat\phi$ would not be diagonal and the loss calculation would need to take this into account.
>
> This type of approach is better suited for cases when the source term in the PDE is an input to the model. For example, something like $\nabla^2\phi(x) = f(x)$, in which case the solution does in fact have that $\frac{\delta\phi(x)}{\delta f(y)} = G(x - y) = \frac{1}{4\pi \|x - y\|} \ne 0$, $G$ being the Green’s function. We leave this case to future work.
>
> Regarding similarity of nearby points, this is captured implicitly by the smoothness of the network.

---

### Official Review · Reviewer_L8FG · 2023-07-07

**Soundness:** 2 fair
**Presentation:** 2 fair
**Contribution:** 2 fair
**Rating:** 3
**Confidence:** 2

**Summary:**

The article describes a method to obtain the solution of a PDE given just the boundary mesh as an input. A variety of edge features are first transformed using MeshCNN, which are then used with a Transformer decoder to obtain the solution. The method is demonstrated to work for a few different PDEs on relatively simple domains.

**Strengths:**

The proposed framework is potentially powerful considering it is supposed to work off of the mesh manifold, which is typically the starting point of the computational analyses.

**Weaknesses:**

The method is demonstrated to work qualitatively, and only on toy problems.
The framework presentation should be improved to demonstrate the framework architecture and how one goes from input (boundary mesh) to output (solution at a physical point in the domain).. I have gone through the text several times and it is still not clear to me. While the authors refer to other papers, I think it is important to make the article self-sufficient to an extent for the reader.
Without sufficient evidence, it is hard to see how this framework would work for complex domains which require large number of elements to accurately solve the PDE of interest.

**Questions:**

It is not clear to me how one goes from the input (mesh manifold) to the solution at any physical point in the domain.

**Limitations:**

It is not clear whether the method can be effective for large scale problems (think 10k+ elements) where the benefits of a fast solution estimator would actually be helpful.

---

> ### Author Rebuttal · Authors · 2023-08-10
>
> Please see our other response and the attached diagram for an explanation of our architecture. We hope this provides a clearer picture of our model.
>
> Regarding the complexity of the geometry, it is important to state that the quality of predictions will be highly dependent on the size of the model and the dataset being trained on. This is no different from image processing problems using CNNs. We expect that, with a high quality dataset and sufficient compute, it should in theory be possible to train a MeshCNN on complex geometry with high resolution detail. However, due to limited compute budget, we are not able to scale up our model at this current time.
>
> Another important point is that we are only considering the boundary geometry, i.e. a closed 2 dimensional surface. This is in contrast to FEM methods and the like which operate on a 3 dimensional volumetric (eg. tetrahedral) mesh. Volumetric meshes require orders of magnitude mode elements for a similar resolution than their corresponding boundary surfaces. A boundary mesh may only need one or two thousand edges to represent very complex geometry, while the interior would require tens or hundreds of thousands of elements for an accurate FEM simulation.

---

> > ### Comment · Reviewer_L8FG · 2023-08-18
> >
> > I appreciate the authors detailed description of the architecture, and I agree with them regarding the comment about scale of the problem size for a 2D surface mesh in contrast to a 3D volume mesh.
> >
> > I still think that demonstrating the impact/practicality/need of any method is crucial, and showing the method working on large scale problems of practical interest is one way of doing that.

---

### Author Rebuttal · Authors · 2023-08-10

# Response to all Reviewers
We kindly thank all reviewers for their time and helpful feedback on our paper. All reviewers agree that our description of our model architecture was confusing and unclear. We aim to give a more concrete and precise description below, which we hope will improve the clarity of our paper. We have furthermore included a figure to give a visual representation. We will of course include this in the final version of our paper.

## Network architecture
The model has two inputs: (1) the boundary geometry in the form of a triangular mesh, and (2) a point $x$ at which the function is being evaluated. These inputs are passed into two subnetworks. See the attached pdf for a diagram of this architecture.

The first subnetwork is the MeshCNN, consisting of alternating MeshConv layers and nonlinear activation functions. The triangular mesh is passed into this network as a tensor of shape `(n_edges, n_features)`, along with an integer index array of shape `(n_edges, 4)` representing the adjacency lists of each edge; each edge having 4 neighbors. The output of the MeshCNN is a tensor of shape `(n_edges, d_embedding)` considered as a latent representation of the boundary geometry. MeshConv layers are able to extract local geometric information about each edge and its neighborhood.

The second subnetwork is based on a Transformer decoder architecture. It is this network which is the PINN, i.e. the one which represents the function $\phi$ in the PDE. This is trained as a PINN conditioned on the geometric information encoded by the MeshCNN. The first layer of this subnetwork is a fully-connected feed-forward network with 1 hidden layer. The input to this layer is the point $x$ which is a 3 dimensional vector. The outputs from this first layer are then fed through a sequence of Transformer blocks. Each block consists of a cross attention layer, followed by another fully-connected feed-forward layer, each with residual connections and layer normalization. The cross attention layers are calculated as follows: $h_\mathrm{out} = \operatorname{attn}(K,V,Q)$, where $K$ and $V$ are linear projections of the edge embeddings (the output of the MeshCNN), and $Q$ is a linear projection of the output from the previous hidden layer (a latent embedding of the point $x$). The softmax and weighted average in the attention operation are computed by summing over the edges. The output of the final Transformer block is then linearly projected to obtain $\phi(x)$.
The purpose of the attention layers is to allow the model to condition on relevant geometric information over the entire mesh.

Here we have described the model assuming that a single mesh and a single point are given as inputs. Of course these can be given in batches. During each training step, we pass to the model a single mesh, and a batch of points sampled within the domain of the PDE. In analogy to typical sequence models, one might say that these points are a batch of length 1 sequences, whilst a single mesh would be analogous to a single sequence (a batch of size 1). In this way, attention weights are computed between each point-edge pair. These weights are only averaged over the edges (the sequence dimension) and not over points (the batch dimension). For each point, the network conditions on information from each edge, but information is not shared between points. If we were to allow information to be shared across the batch dimension, it would break underlying assumptions of the system being solved. For example, consider two points $x$, $y$ in the domain. We have that $\frac{\partial x}{\partial y} = 0$, since these are independent points. However, if we allow information to be shared across the batch dimension, we would have $0 \ne \frac{\partial \phi(x)}{\partial y} = \frac{\partial\phi(x)}{\partial x} \frac{\partial x}{\partial y} = 0$, which is nonsensical. It is of course possible to build a model which attends over many points in the domain; however, this type of model would need to be trained differently to correct for this mixing.

---

### Author Response · Authors · 2023-08-21

We would like to once again thank all the reviewers for their feedback and comments.
This feedback has been informative for us, and will certainly help us improve the clarity and quality of our paper for the next revision.

---

### Decision · Program_Chairs · 2023-09-21

**Decision:**

Accept (poster)

**Comment:**

The reviews are somewhat conflicting for this paper, but the AC gave it a read and decided to suggest accepting this work.  The idea of learning a PDE solution as a function of boundary mesh geometry is a creative and interesting one, so the AC is willing to overlook some weaknesses (most importantly, limited stress testing).

In the final revision, please address the reviewer comments, which were thoughtful and reasonable.  In particular, the exposition needs to be revised to include clear descriptions of the input/output, architecture, and hyperparameters.  Optionally, "stress-testing" by trying more complicated boundary geometries or PDEs would place the paper on more rigorous footing.